# Total Plasma Exchange in Neuromuscular Junction Disorders—A Single-Center, Retrospective Analysis of the Efficacy, Safety and Potential Diagnostic Properties in Doubtful Diagnosis

**DOI:** 10.3390/jcm11154383

**Published:** 2022-07-28

**Authors:** Andreas Totzeck, Michael Jahn, Benjamin Stolte, Andreas Thimm, Christoph Kleinschnitz, Tim Hagenacker

**Affiliations:** 1Department of Neurology and Center for Translational Neuro- and Behavioral Sciences (C-TNBS), University Hospital Essen, University of Duisburg-Essen, 45147 Essen, Germany; benjamin.stolte@uk-essen.de (B.S.); andreas.thimm@uk-essen.de (A.T.); christoph.kleinschnitz@uk-essen.de (C.K.); tim.hagenacker@uk-essen.de (T.H.); 2Department of Nephrology, University Hospital Essen, University of Duisburg-Essen, 45147 Essen, Germany; michael.jahn@uk-essen.de

**Keywords:** myasthenia gravis, Lambert–Eaton syndrome, decrement, MGFA, QMG

## Abstract

Neuromuscular junction disorders (NJDs) are a heterogeneous group of diseases including myasthenia gravis (MG). In some cases, patients are present with myasthenic symptoms without evidence of autoimmune antibodies, making diagnosis challenging. Total plasma exchange (TPE) has proven efficacy in NJDs. The objective is to describe the safety and efficacy of TPE in NJD patients with questionable disease activity or uncertain diagnosis in order to assess the diagnostic potential of TPE. We report an observational, retrospective cohort study of clinical routine data. All the data were derived from the electronic medical records of the Department of Neurology at University Hospital Essen. We searched for patients with NJDs between 1 July 2018 and 30 June 2021. Of the 303 patients who presented to the department with NJDs, 20 were treated with TPE; 9 patients did not show a measurable benefit from TPE (45%), 6 of whom were diagnosed with seronegative MG. Of these, 3 (50%) had long-standing ocular symptoms. There were decreases in the mean arterial pressure, hemoglobin, hematocrit and fibrinogen during treatment, which were not considered clinically relevant. In (seronegative) myasthenic patients, TPE may help to verify an uncertain diagnosis or to reveal possible muscle damage, allowing unnecessary therapy to be avoided.

## 1. Introduction

Neuromuscular junction disorders (NJDs) are a heterogenous disease group that typically involve exercise-dependent weakness in the ocular and proximal muscles of the limb and trunk, including in bulbar and ventilatory function [1,2]. Myasthenia gravis (MG) is the main representative of this group and is a chronic autoimmune disease that involves the dysfunction of neuromuscular junctions mediated by antibodies against proteins of the postsynaptic membrane, including the acetylcholine receptor (ACh-R) [3], muscle-specific kinase (MuSK) [4] and low-density lipoprotein receptor-related protein 4 (LRP4) [5]. By contrast, Lambert–Eaton myasthenic syndrome (LEMS) is a presynaptic antibody-mediated disorder that is often associated with cancer, involving autoantibodies against presynaptic P/Q-type voltage-gated calcium channels (VGCCs) [6]. However, in some cases, antibody results remain negative, impeding diagnosis. Patients with such a doubtful diagnosis are typically diagnosed as seronegative and treated similarly to antibody-positive patients. Corticosteroids, symptomatic drugs such as cholinesterase inhibitors (C-I) or 3,4-diaminopyridine (3,4-DAP), and steroid-sparing immunosuppressive drugs remain the basic therapy [7]. These all have different side effects that accumulate during life-long therapy; therefore, particularly in seronegative patients, the question of diagnostic certainty arises. In patients with a confirmed NJD diagnosis, it may be uncertain whether the disease is still active or whether damage has already occurred. A tool for NJD would, therefore, be helpful for distinguishing between active disease or chronic muscle damage, analogously with other autoimmune diseases [8,9], in order to determine whether escalation therapy or other appropriate treatments should be provided. In the case of rapid progressive disease, a refractory response or critical deterioration, e.g., myasthenic crisis with ventilatory dysfunction, treatment with intravenous immunoglobulin (IVIg) or total plasma exchange (TPE), a method of extracorporeal blood purification with the intention of removing inflammatory mediators and antibodies, is an effective therapeutic option [10]. Patients with moderate-to-severe antibody-positive MG and non-responsiveness to immunosuppressive agents have benefited from periodic TPE treatment [11]. Although more invasive than IVIg, TPE is considered a fast and safe method in critically ill patients and has been applied to patients with different autoimmune diseases involving rapid deterioration [12]. Seronegative MG patients more often present with milder or ocular symptoms compared to antibody-positive MG patients, although TPE is equally effective in patients with critical deterioration [13,14,15]. However, seronegative patients with myasthenic crisis are typically young, and the diagnosis of the crisis is frequently delayed [13]. In this non-confirmatory, exploratory analysis, our objective was to determine the safety and efficacy of TPE in a cohort of NJD patients with uncertain status of disease activity or with questionable diagnosis, in order to assess the potential diagnostic role of TPE.

## 2. Materials and Methods

### 2.1. Study

We report a retrospective, observational cohort study conducted at the Department of Neurology, University of Duisburg-Essen, Germany. The study was conducted in accordance with the Declaration of Helsinki and approved by the Institutional Ethics Committee of University Hospital Essen (protocol code: 22-10707-BO; date of approval: 10 May 2022). Patient consent was waived due to the retrospective design of the study with completely anonymized data analysis.

### 2.2. Patients

The clinical database was retrospectively searched for patients who were treated at the tertiary center for neuromuscular diseases in the Department of Neurology, University of Duisburg-Essen, Germany. The data collection period was set between 1st July 2018 and 30th June 2021. Patients who were diagnosed in accordance with the International Statistical Classification of Diseases and Related Health Problems (ICD) G70.0–G70.9 or G73.0–G73.3 were included. Diagnosis was based on the patient’s medical history; clinical characteristics, such as the involved muscle groups, fluctuating fatigability and muscle weakness; a positive response to cholinesterase inhibitors; and, optionally, a recorded decrement in repetitive motor nerve stimulation [1]. The detection of autoimmune antibodies (i.e., antibodies against AChR, LRP4, MuSK or P/Q-type channels) affirmed a diagnosis of seropositive MG/LEMS. MG and its clinical subtypes were diagnosed according to the Myasthenia Gravis Foundation of America (MGFA) classification. The Quantitative Myasthenia Gravis Score (QMG score) was measured [16]. In a second step, we explored how many of these patients had previously received TPE or immunoadsorption (IA), the benefit of which was assessed according to inpatient medical records.

### 2.3. Plasma Exchange

Plasma exchange therapy was routinely performed according to the standard operating procedure. For TPE, the centrifugal devices Spectra Optia (Terumo BCT, Inc., Lakewood, CO, USA) and COM.TEC (Fresenius, Bad Homburg, Germany) were used. For the anticoagulation of the extracorporeal circuit, a 3% acid citrate dextrose solution, USP Formula A (ACD-A) (Haemonetics, Munich, Germany), was added to the tubing sets during the apheresis procedures, with a regular anticoagulation ratio of 1:20. The plasma volume was estimated with the Kaplan formula (estimated plasma volume = (0.065 × weight [kg]) × (1-hematocrite)), and the maximal plasma volume was limited to 4 L [17]. We adopted the TPE strategies described in previous publications and guidelines exchanging the simple plasma volume with 4% albumin [18,19,20,21]. For IA, the plasma separator Plasmaflo OP-08W and the plasma adsorber Immusorba TR-350 (both Asahi Kasei Medical, Tokyo, Japan) were applied via the plasma exchange monitor Plasauto Sigma (Asahi Kasei Medical, Tokyo, Japan) or OctoNova (Diamed, Cologne, Germany). The anticoagulation of the extracorporeal circuit consisted of the injection of heparin as a bolus (Median 3500 IE) prior to the therapy and the continuous infusion of heparin (median 1500 IE/h) throughout the IA. According to the manufacturer’s data and treatment protocols of previous studies, the treated plasma volume was 2000 mL with a plasma flow of 20 mL/min [22,23]. The treatment consisted of 5 TPE or IA sessions every second day utilizing a double-lumen catheter placed into the jugular vein. Blood pressures were taken before, during and after each TPE or IA procedure.

### 2.4. Laboratory Analysis

Routine blood samples were taken before and after the TPE treatments. Each sample was directly examined for blood cell counts as well as hemoglobin, hematocrit and fibrinogen. The blood samples were analyzed in the laboratory of University Hospital Essen, Germany. The reference range was defined and established by the laboratory by comparison with the normal population.

### 2.5. Statistics

For descriptive statistics, continuous variables are reported as the means and standard errors of the means. The mean values and SEMs of the patients’ characteristics and laboratory results were calculated using PRISM 8 (GraphPad Software, San Diego, CA, USA). The mean age of all the patients and that of the patients receiving TPE were compared via non-parametric Mann–Whitney tests. The blood pressure and laboratory parameters were paired for each patient, and the changes were analyzed with paired *t*-tests and ANOVA. *p* < 0.05 was considered significant.

## 3. Results

### 3.1. Diagnosis and Indication

During the analyzed period of three years, 303 patients presented with myasthenic symptoms. Out of these 303 patients, 20 received TPE (6.6%) and 4 (1.3%) received IA (Figure 1 and Table 1). One half were male, and the other half were female patients (n = 12 vs. n = 12). The mean age was 63.8 ± 3.5 years. The youngest patient was 24, and the oldest was 87 years old. Patients receiving TPE or IA were older than the patients (n = 303) presenting with myasthenic symptoms (*p* = 0.0167).

Due to the small number of IA patients (n = 4), only TPE patients were further evaluated (Table 2). Eleven patients benefitted from TPE according to patient records (55.0%). Out of these eleven patients, there were nine patients (Patients 10–18) with acute deterioration or myasthenic crisis of ACh-R antibody-positive MG, two patients requiring mechanical ventilation, and two patients with LEMS (Patients 19 and 20). Only one seronegative patient responded to TPE treatment. One patient partially improved (Patient 9) and continued to be dependent on mechanical ventilation due to intrinsic pulmonary disease. Out of the eight patients who did not improve, seven were seronegative. Patient 8 was the only individual with ACh-R-antibody-positive MG who did not benefit from TPE, assuming a state of persistent muscle damage after long-lasting juvenile MG. Patients 1 and 2 had no MG and MG medication was stopped, respectively. Patient 2 was later treated for blepharospasm. Patients 3 and 4 were diagnosed with seronegative ocular MG. Due to the lack of improvement after TPE, there was no escalation of immunotherapy. Patients were later presented to an ophthalmologist. Patients 5, 6 and 7 were diagnosed with seronegative MG. The clinical deterioration in these patients was due to underlying comorbidities (hypertrophic obstructive cardiomyopathy, chronic ischemic heart failure, and hyperparathyroidism, respectively). Thus, there was, again, no escalation of MG therapy for these patients. Among patients who benefited from TPE, arterial hypertension (82%), pneumonia (55%), hypothyroidism, diabetes, and chronic back pain (all 36%) were the most common comorbidities.

### 3.2. Safety

The mean arterial pressure (MAP) was 89.1 ± 1.1 mm Hg before and 87.0 ± 1.1 mm Hg after the TPE procedure. The maximum MAP was 92.8 ± 1.2 mm Hg, and the minimum MAP was 80.5 ± 1.0 mm Hg during the TPE procedure. There was a significant decline in MAP during the TPE procedure compared with the initial MAP (*p* < 0.0001). The highest blood pressure of one patient before the start of TPE was 205/86 mm Hg. The lowest mean arterial pressure patient during the TPE procedure was 61 mm Hg and occurred only in one patient. None of the patients required higher levels of catecholamines due to the TPE procedure. The fibrinogen levels decreased during the five TPE treatments, from 361.8 ± 32.5 to 145.3 ± 5.5 mg/dL (*p* < 0.0001, Figure 2). There was no significant reduction in white blood cell (WBC) count and platelets (9.38 ± 1.03 vs. 9.26 ± 0.70/nL, *p* = 0.8791; 241.2 ± 14.0 vs. 243.8 ± 16.3/nL, *p* = 0.8724; Figure 3A,B). However, the hemoglobin levels and hematocrit after TPE treatment were lower than the levels before the initiation of therapy (12.71 ± 0.53 vs. 11.31 ± 0.47 g/dL, *p* = 0.0002; 0.38 ± 0.02 vs. 0.34 ± 0.01 L/L, *p* = 0.0010; Figure 3C,D). None of the patients needed a blood transfusion due to the TPE procedure.

## 4. Discussion

### 4.1. Patient Characteristics

This retrospective study aimed to evaluate the use of TPE in a cohort of patients with NJDs, and its safety as part of a daily routine and its indications were assessed. Fewer than 10% of the patients with NJD were treated with TPE or IA. This is comparable to the situation for a German cohort of MG patients (n = 1660), in which 7.2% were treated with TPE or IA as part of therapeutic escalation [24]. The composition of our cohort differed from the usual distribution spectrum for myasthenia due to the tertiary center setting [25]. The patients receiving TPE were older, and the majority were diagnosed with worsening or myasthenic crisis of ACh-R-antibody-positive generalized MG. The TPE for MG was first described by Pinching and Peter in three patients in 1976 [26]. TPE directly removes ACh-R antibodies from the circulation, and the clinical and functional outcomes correlate with a decline in antibody levels [27,28,29]. Along with the removal of antibodies, a reduction in inflammatory mediators, such as cytokines and complement factors, induced by TPE plays an important role [30,31,32]. Although TPE and IVIg are equally effective in treating myasthenia gravis [33], it appears that the clinical response to TPE is often more rapid than that to IVIg as part of a daily routine. A recent meta-analysis revealed a higher response rate with TPE than IVIg in acute MG patients [34]. The beneficial effect of TPE can be observed within days [35]. The evaluation time during hospitalization in our study was, therefore, long enough for assessing clinical improvement. All the patients received five TPE treatments, which should be sufficient for evaluating any possible effects. A mean maximum 71.0% reduction in ACh-R-autoantibody titers was observed after 5–6 TPE procedures in MG patients [28]. After a single IA procedure, a reduction of 52.5% in AChR-autoantibody titers was observed [23]. In our study, almost all the patients with ACh-R-antibody-positive MG benefitted from TPE (91%), and 55% of all the treated patients clinically improved. Ebadi et al. reported that TPE was effective in 57% of patients, but ACh-R-autoantibody positivity was not an independent predictor [36]; thus, the underlying type of antibody does not play a role. Likewise, the antibody titer does not correlate with the severity of the disease [37]. Individually, an antibody overshoot weeks after TPE may lead to clinical deterioration in MG patients [38]. Ebadi et al. collected QMG scores to evaluate the treatment’s effectivity [36].

Two patients with LEMS also improved after TPE. TPE is a possible alternative for the acute management of LEMS [39]. Case series have stated improvements in clinical and electrophysiological outcome measures, showing at least a transient decrease in the serum levels of anti-VGCC [6,40]. To date, there have been no systematic studies on TPE for LEMS patients.

### 4.2. Diagnostic Value of TPE

MG is a chronic disease often requiring life-long treatment after diagnosis [1,2,7]. In some cases, making a diagnosis can be difficult, even in specialized centers. Patients, presented to numerous experts in advance, may reveal incongruent electrophysiological or clinical findings, or present as seronegative after repeated tests. Although antibody testing has improved and new antibodies have been discovered, leading to a decline in seronegative MG cases [41,42], there are still commonly patients with a myasthenic syndrome and repeated negative test results, potentially indicating the existence of unknown antibodies or antibodies not detectable by current research methods. These patients may show slight improvement under steroid or immunosuppressive therapy; thus, the escalation of immunosuppression will inevitably be discussed. In other autoimmune disorders, clear markers and scores exist for differentiating between disease activity or damage that has already occurred. In dermatomyositis, an autoimmune disorder that primarily affects the skin, muscles and lungs, tools such as the Myositis Damage Index, the Myositis Disease Activity Assessment Tool and the Disease Activity Score provide important information on the disease activity and, thus, the required level of treatment [9]. No equivalent testing strategies for patients with MG have yet been designed. We hypothesize that TPE may help in deciding if the escalation of immunosuppressive therapy is reasonable when other diagnostic options have failed. Usually, TPE is not a diagnostic procedure. Although TPE seems equally effective in seronegative MG patients with critical deterioration [13], in our cohort, seven patients with seronegative myasthenic syndrome did not satisfactorily benefit from the TPE procedure, resulting in no further escalation of immunosuppressive therapy. Clinical follow-up revealed a different diagnosis in two patients mimicking myasthenic syndromes (blepharospasm and frailty with chronic pulmonary disease). Ocular symptoms, in particular, did not improve with TPE. The histopathological findings in myasthenic extraocular muscles showed lymphocytic infiltrates in the MG patients, with a disease duration of less than 3 years on average [43,44]. In two MG patients with total external ophthalmoplegia, extraocular muscles were almost completely replaced with fat [43]. Similar fibrofatty changes in extraocular muscles were also found in MG patients with treatment-resistant ophthalmoplegia in ACh-R-antibody-positive MG after 1.2 years, and seronegative MG after 3.5 years [45,46]. This implies early muscle damage in ocular MG patients, which would explain why these patients are refractory to therapy. This muscle damage may also occur in long-standing generalized ACh-R-antibody-positive MG. In our cohort, a patient with juvenile generalized ACh-R-antibody-positive MG did not improve after TPE treatment (Patient 8). Several immunosuppressive therapies had been applied with unsatisfactory reductions in myasthenic symptoms in previous years for this patient. Nevertheless, eculizumab, a complement inhibitor for patients with ACh-R-antibody-positive refractory generalized MG [47], was not tested in this patient. Prospective studies are needed to clarify if TPE is capable of demasking possible muscle damage in MG patients to avoid unnecessary therapeutic escalation at an early stage, especially in elderly NJD patients, with comorbidities leading to potential drug interactions and side effects from long-term immunosuppression.

### 4.3. Safety of TPE

The TPE procedure was able to be performed safely in a cohort of inpatients with NJDs at a tertiary center of neuromuscular diseases. Hypotension was defined as an MAP below 65 mm Hg [48]. Even in elderly patients with comorbidities, adequate blood pressure was easily maintained during the TPE procedure. Szczeklik et al. treated 54 ICU patients with TPE, 18 (33.3%) of which presented with MG [12]. In 7.3% of the TPE procedures, non-life-threatening falls in arterial blood pressure that did not require catecholamines occurred. Nevertheless, in about 1% of the TPE procedures, additional pressor amines were required [12]. In a study with 42 MG patients receiving TPE, only one patient was reported with hypotension as a severe complication [36]. Our study also revealed a decline in MAP during the TPE procedure, without any need for additional catecholamines; only one patient had hypotension as defined.

Although the WBCs and platelets were stable, there was a decrease in hemoglobin and hematocrit, indicating a possible intravasal volume overload. However, hemolysis caused by TPE may also have been a cause of this decrease. Szczeklik et al. reported one patient with hemolysis under TPE treatment in their cohort of ICU patients [12]. The prevention of hemolysis in the extracorporeal circulation of a patient’s blood has always been an important issue. Obstacles, such as hemolysis, can be technologically overcome by, for example, using special membranes (hollow fiber) with a small surface area or by monitoring the transmembrane pressure [49,50]. Due to the retrospective design of this study, there were no sufficient data on haptoglobin, fragmentocytes or lactate dehydrogenase to analyze. Nonetheless, mild hemolysis of red blood cells was also observed in patients who received IVIg but no TPE [51].

The decrease in fibrinogen during TPE treatment is very well known and is possibly an indirect indication of whether the procedure works. In our opinion, the individual decline does not correlate with a therapeutic effect; however, further studies will be necessary. The careful monitoring of fibrinogen, hemoglobin and hematocrit is important, especially for anemic patients or patients at a high risk of bleeding.

### 4.4. Limitations

Limitations are inherent to single-center, retrospective analyses. In our study, there was no prospective control for confounders, but all the data were derived from the EHR. This limitation was mitigated because we adhered to the standard operating procedures and used reproducible and reliable measurements. However, study-related measures, such as additional blood work analysis, were not possible. In addition, the QMG scores for patients after TPE treatment were missing; these scores would have provided a more advantageous option for evaluating the clinical benefit of the treatment. Ultimately, the case number is low, which means that the study can be considered beneficial in generating a hypothesis rather than being confirmatory.

## 5. Conclusions

This study justifies conducting a prospective trial, evaluating whether TPE can help to confirm or even reject a diagnosis in patients with myasthenic symptoms. In MG patients, TPE may assist in avoiding the further escalation of immunosuppressive treatment by addressing the question of muscle damage. With the inherent limitation of a monocentric, retrospective analysis, TPE can be considered a safe procedure, even in elderly patients with NJD.

## Figures and Tables

**Figure 1 jcm-11-04383-f001:**
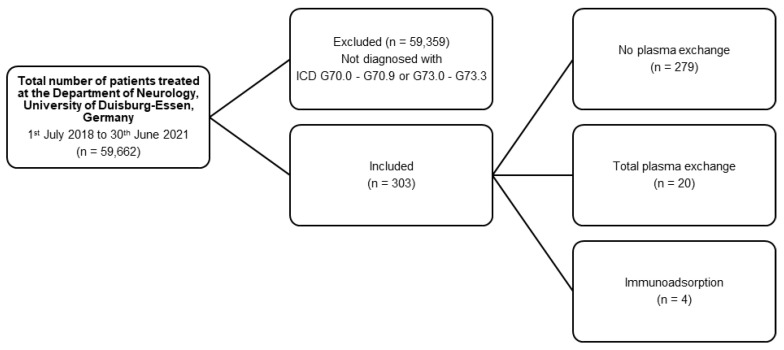
Flow diagram of the retrospective analysis.

**Figure 2 jcm-11-04383-f002:**
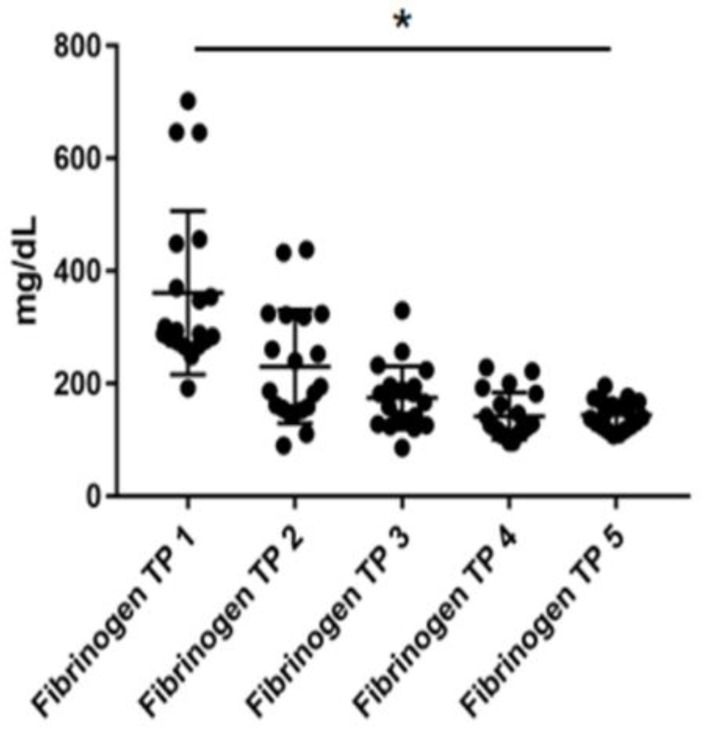
Levels of fibrinogen at five different timepoints (TP) during total plasma exchange procedure. * = *p* < 0.05. Significant difference between TP 1 and every following TP.

**Figure 3 jcm-11-04383-f003:**
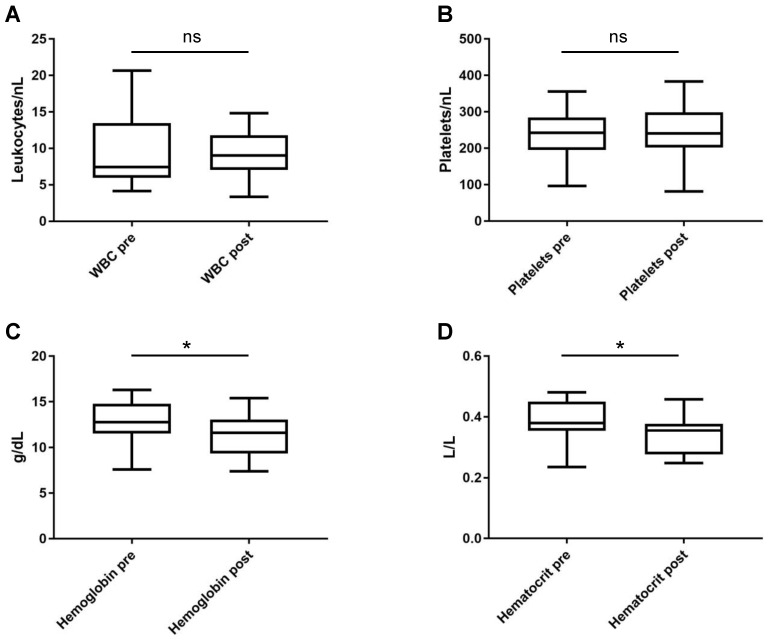
Levels of (**A**) white blood count (WBC), (**B**) platelets, (**C**) hemoglobin, and (**D**) hematocrit, before (pre) and after (post) total plasma exchange. Boxplots show medians and minima to maxima. ns = not significant. * = *p* < 0.05.

**Table 1 jcm-11-04383-t001:** Demographics and ICD codes of patients presenting with neuromuscular junction disorders (n = 303).

Demographics	
Mean age in years ± SEM, (range)	53.2 ± 1.2 (14–91)
Male	46,9% (n = 142)
Female	53.1% (n = 161)
**ICD [n]**	
G70.0—Myasthenia gravis	252
G70.1—Toxic myoneural disorders	3
G70.2—Congenital and developmental myasthenia	8
G70.8—Other specified myoneural disorders	10
G70.9—Myoneural disorder, unspecified	26
G73.0—Myasthenic syndromes in endocrine diseases	0
G73.1—Lambert–Eaton syndrome in neoplastic disease	2
G73.2—Other myasthenic syndromes in neoplastic disease	2
G73.3—Myasthenic syndromes in other diseases classified elsewhere	0

**Table 2 jcm-11-04383-t002:** Patients who were treated with total plasma exchange. f = female; m = male; MGFA = Myasthenia Gravis Foundation of America classification; QMG = Quantitative Myasthenia Gravis score; CH-I = cholinesterase inhibitors; 3,4-DAP = 3,4-diaminopyridine; AZA = azathioprine; MMF = mycophenolate mofetil; IVIG = intravenous immunoglobulin; RITUX = rituximab; ECU = eculizumab; ACh-R = acetylcholine receptor (antibodies); MUSK = muscle-specific kinase (antibodies); LRP4 = low-density lipoprotein receptor-related protein 4 (antibodies); P/Q or N type (voltage-gated calcium channel antibodies); Titin (antibodies); MG = myasthenia gravis; LEMS = Lambert–Eaton myasthenic syndrome; COPD = chronic obstructive pulmonary disease.

Patient [no.]	Gender	Age [y]	Decrement	MGFA	QMG	Steroids	CH-I	3,4-DAP	AZA	MMF	IVIg	RITUX	ECU	ACh-R	MuSK	LRP4	P/Q type	N type	Titin	Thymoma	Improvement	Diagnosis
1	f	74	-	n/a	n/a	+	+	-	-	-	-	-	-	-	-	-	-	-	-	-	-	COPD/Frailty
2	f	63	+	I	3	+	+	-	+	-	-	-	-	-	-	-	-	-	-	-	-	Blepharospasm
3	f	78	-	I	22	+	+	-	-	-	-	-	-	-	-	-	-	-	-	-	-	MG
4	m	41	-	I	3	+	+	-	+	-	-	-	-	-	-	-	-	-	-	-	-	MG
5	f	75	-	IIIA	13	+	+	-	+	-	-	-	-	-	-	-	-	-	-	-	-	MG
6	m	58	-	IIB	12	+	+	-	-	-	-	-	-	-	-	-	-	-	-	-	-	MG
7	m	81	-	V	17	+	+	-	-	+	-	-	-	-	-	-	-	-	-	-	-	MG
8	f	24	+	IVB	11	+	+	-	+	+	+	+	-	+	-	-	-	-	-	-	-	MG
9	f	78	-	V	22	+	+	-	-	-	-	+	-	+	-	-	-	-	+	-	(+)	MG
10	m	84	+	V	17	+	+	-	+	-	-	-	-	+	-	-	-	-	-	-	+	MG
11	m	81	+	IVB	23	+	+	-	+	-	-	-	-	+	-	-	-	-	+	-	+	MG
12	m	76	-	IIIB	7	+	+	-	-	+	-	-	-	+	-	-	-	-	-	-	+	MG
13	f	87	+	IVB	23	+	+	-	-	+	-	-	-	+	-	-	-	-	-	-	+	MG
14	m	64	-	IVB	10	+	+	-	+	-	-	-	-	+	-	-	-	+	+	-	+	MG
15	m	38	+	IVB	9	+	+	-	-	+	-	-	-	+	-	-	-	-	-	+	+	MG
16	f	41	+	IVB	24	+	+	-	+	-	-	-	+	+	-	-	-	-	-	-	+	MG
17	f	77	+	IVB	14	+	+	-	+	-	-	-	-	+	-	-	-	-	+	-	+	MG
18	m	59	+	V	28	+	+	-	-	-	+	-	+	+	-	-	-	-	+	-	+	MG
19	m	77	-	n/a	n/a	+	-	+	+	-	+	-	-	-	-	-	-	-	-	-	+	LEMS
20	m	54	-	n/a	6	+	-	+	+	-	+	-	-	-	-	-	+	-	-	-	+	LEMS

## Data Availability

The data presented in this study are available in the article.

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
