# Peer review of "Total Plasma Exchange in Neuromuscular Junction Disorders—A Single-Center, Retrospective Analysis of the Efficacy, Safety and Potential Diagnostic Properties in Doubtful Diagnosis"

_jcm, 2022, doi:10.3390/jcm11154383_

Round 1
Reviewer 1 Report
[JCM] Manuscript ID: jcm-1819785
Total plasma exchange in neuromuscular junction disorders — a single-center, retrospective analysis of the efficacy, safety and potential diagnostic properties in doubtful diagnosis
It should be added to the discussion that PE is not a means to falsify diagnoses.
If a pat is seronegative there are no antibodies that could be removed by PE or antibodies that have not been identified yet.
The English should be improved
7/22
Author Response
- It should be added to the discussion that PE is not a means to falsify diagnoses. We thank the reviewer for the remark. A sentence has been added to the discussion section (line 258).
- If a pat is seronegative there are no antibodies that could be removed by PE or antibodies that have not been identified yet. The issue was addressed in the introduction as well as in the discussion section. A sentence was added to the discussion to clarify the point. (Lines 258-259)
- The English should be improved Extensive English editing was performed by MDPI author services to improve the English language.
Reviewer 2 Report
Totzeck and colleagues present an interesting retrospective exploratory study to determine the safety and efficacy of total plasma exchange (TPE) treatment in NDJ patients with uncertain status.
Although, the number of cases was low (n=20), collected data are sufficient, as authors mention, to generate a hypothesis, and justifies the execution of a future prospective trial.
Only some minor points I consider should be addressed:
1.-Is it possible to include references on QMG score and plasma exchange protocol in the Material and Methods section?
2.-Fibrinogen levels decreased at different time points of TPE treatment (Lines 169-170 and Figure 2): Is there a significant difference among all the time points? Significant difference corresponds only to TP1 vs TP5? Please, clarify this point.
3.-Define WBC at the first time it is mentioned
4.-Please correct units on Figure 3A and 3B
Author Response
- Is it possible to include references on QMG score and plasma exchange protocol in the Material and Methods section? We thank the reviewer for the helpful remarks. References on QMG score and TPE protocols were added to the Methods section.
- Fibrinogen levels decreased at different time points of TPE treatment (Lines 169-170 and Figure 2): Is there a significant difference among all the time points? Significant difference corresponds only to TP1 vs TP5? Please, clarify this point. There was a significant difference between TP 1 and every following TP. The missing information was added (line 195).
- Define WBC at the first time it is mentioned The definition of WBC was added in line 187.
- Please correct units on Figure 3A and 3B The units of Figure 3A and 3B were corrected.
Reviewer 3 Report
Overall, this is a very explanatory and well written article. The topic is very important and it will aid to better understand a very difficult illness as MG is. The introduction is well written and gives enough information. The methods are clearly explained and provide enough information.
However, in the results more data should be provided. There are some results that are discussed but are not reported in the results section.
To finish, the discussion and conclusions are well written and only small corrections should be performed.

Author Response
- Please, include TPE in the keywords
- We thank the reviewer for this helpful remark. TPE has been added to the keywords.
- Could you please report if the patients who responded to the treatment had any comorbidities?
- Data on comorbidities of the responder group was added to the results section (lines 160 – 162).
- Please explain how many that responded to the treatment were seronegative.
- Only one seronegative LEMS patient responded to the treatment. A sentence was added to the results section (lines 147 – 148).
- Could you please include and relate the Ach-R levels during and after the treatment?
- We thank the reviewer for the recommendation, however, due to the retrospective analysis, there is no sufficient data on ACh-R antibody levels. Current antibody levels were only available in three patients during hospitalization, but they were also not controlled after TPE. Measuring antibodies over time would be a good approach for a prospective study design.
- There are results discussed which have not been included in the results section: histopathological findings, ACh-R levels, fibrofatty changes in extraocular muscles
- Unfortunately, no muscle biopsies were performed in any of our patients. As also explained under point 4, there were current measurements of ACh-R antibodies in only 3 patients, but these were also not checked directly after TPE.